# Chronic Dehydration in Nursing Home Residents

**DOI:** 10.3390/nu12113562

**Published:** 2020-11-20

**Authors:** Masaaki Nagae, Hiroyuki Umegaki, Joji Onishi, Chi Hsien Huang, Yosuke Yamada, Kazuhisa Watanabe, Hitoshi Komiya, Masafumi Kuzuya

**Affiliations:** Department of Community Healthcare and Geriatrics, Nagoya University Graduate School of Medicine, 65 Tsuruma-cho, Showa-ku, Nagoya, Aichi 466-8550, Japan; m-nagae@med.nagoya-u.ac.jp (M.N.); j-onishi@med.nagoya-u.ac.jp (J.O.); huang.chi.hsien@h.mbox.nagoya-u.ac.jp (C.H.H.); yoyamada@med.nagoya-u.ac.jp (Y.Y.); n910eb@gmail.com (K.W.); h-komiya@med.nagoya-u.ac.jp (H.K.); kuzuya@med.nagoya-u.ac.jp (M.K.)

**Keywords:** chronic dehydration, nursing home, serum osmolality, inferior vena cava, ultrasound, dementia, body mass index, older

## Abstract

Chronic dehydration mainly occurs due to insufficient fluid intake over a lengthy period of time, and nursing home residents are thought to be at high risk for chronic dehydration. However, few studies have investigated chronic dehydration, and new diagnostic methods are needed. Therefore, in this study, we aimed to identify risk factors for chronic dehydration by measuring serum osmolality in nursing home residents and also to evaluate whether examining the inferior vena cava (IVC) and determining the IVC collapsibility index (IVC-CI) by ultrasound can be helpful in the diagnosis of chronic dehydration. A total of 108 Japanese nursing home residents aged ≥65 years were recruited. IVC measurement was performed using a portable handheld ultrasound device. Fifteen residents (16.9%) were classified as having chronic dehydration (serum osmolality ≥295 mOsm/kg). Multivariate logistic regression analysis showed that chronic dehydration was associated with dementia (odds ratio (OR), 6.290; 95% confidential interval (CI), 1.270–31.154) and higher BMI (OR, 1.471; 95% CI, 1.105–1.958) but not with IVC or IVC-CI. Cognitive function and body weight of residents should be considered when establishing a strategy for preventing chronic dehydration in nursing homes.

## 1. Introduction

The adult human body consists of about 60% water, with muscle functioning as the main reservoir of water, but in older adults, this amount is reduced to only around 50% due to reduced muscle mass. Among older adults, dehydration is common and is associated with several serious adverse events, including longer hospital stays and higher mortality rates [1]. Dehydration also imposes substantial financial costs on society. For example, in the United States, Medicare reimbursed over $446 million for hospitalizations due to dehydration in a single year “in 1991” [2,3].

Dehydration can be classified as either acute or chronic. Acute dehydration results from excessive water loss caused by acute illness (e.g., infection), intense physical exercise, or heatstroke. Acute dehydration is considered relatively easy to notice because it is mostly predictable and leads to moderate-to-severe dehydration [4,5]. In contrast, chronic dehydration is mainly due to insufficient fluid intake over a lengthy period and is often insidious [4,5].

Older adults are considered at risk for chronic dehydration due to their reduced sensitivity to thirst, lower urine concentrating ability, and lower fluid intake compared with young or middle-aged adults [5,6]. Furthermore, older nursing home residents may be at high risk for chronic dehydration because lower fluid intake is commonly observed in nursing home residents [7,8].

Systematic reviews of dehydration in nursing home residents have identified many risk factors for dehydration, including sex, diabetes, renal impairment, cognitive impairment, and oral feeding [4,9]. However, most of these studies did not clearly indicate whether acute or chronic dehydration was investigated and the methods for diagnosing dehydration varied between studies. As a result, the current literature on chronic dehydration is insufficient, and the risk factors for chronic dehydration remain to be fully elucidated.

There is no generally accepted gold standard for assessing dehydration in older adults but most studies diagnosed dehydration based on the results of blood tests, including serum sodium, serum osmolality, and the blood urea nitrogen (BUN)-to-creatinine ratio [4]. A previous systematic review recommended serum or plasma osmolality as a reference standard for water-loss dehydration because osmolality is strictly regulated, and any change suggests problems in the body [10]. At present, serum osmolality is considered the best test for diagnosing dehydration in older adults [11] and nursing home residents [12].

Although some studies have reported no difference in serum osmolality between habitually low fluid intake and high fluid intake [13], even a slight increase in serum osmolality was found to stimulate the release of arginine vasopressin and cortisol, which are known to heighten the risk of chronic diseases. In clinically stable long-term care residents, elevations of serum osmolality were frequently observed without clinically overt signs of dehydration [14], so using serum osmolality to assess chronic dehydration is considered reasonable. Chronic dehydration may be more problematic among nursing home residents [7,8], and establishing effective preventive strategies for chronic dehydration in this population would require identifying the risk factors for chronic dehydration.

Recently, increasing attention has been focused on point-of-care ultrasonography (POCUS), which is a type of ultrasonography performed by clinicians as part of medical treatment [15]. POCUS can be used in situations both inside and outside the hospital, and because POCUS images can be obtained immediately, and scans can be repeated easily; clinicians can assess the patient’s condition in real time. With regard to the cardiovascular system, the American Society of Echocardiography used the term focused cardiac ultrasound (FCU). FCU is a focused examination of the cardiovascular system performed by the physician completing the physical examination [16]. Some FCU training programs include assessment of volume status by measuring the inferior vena cava (IVC) diameter or IVC collapse.

IVC measurement by ultrasound is commonly used in emergency departments or intensive care units for fluid assessment in suspected cases of dehydration. A meta-analysis of an adult population reported that the maximal IVC diameter of hypovolemic groups was significantly lower than that of euvolemic groups, with a mean difference of 6.3 mm [17]. In contrast, the utility of IVC for evaluating dehydration in older patients remains controversial. Two studies reported a significantly lower maximal IVC diameter in hypovolemic groups compared with euvolemic groups, with a mean difference of 4 mm [18,19], whereas another study reported no difference in IVC diameter between the two groups [20]. One reason for this disparity may be that these studies used the BUN/Cr ratio or clinical signs as a surrogate marker of dehydration rather than serum osmolality.

It is not always easy for nursing home residents to undergo medical examinations because many of them have physical disabilities or limitations in activities of daily living (ADL). Measurement of IVC using ultrasound at bedside is non-invasive, repeatable, and does not require special techniques, and it may also be useful outside the hospital setting, especially in nursing homes. If measurement of IVC by portable ultrasound is deemed useful in diagnosing chronic dehydration, clinicians could perform the assessment quickly, potentially avoid complications and hospital admissions.

It is expected that by clarifying the risk factors for chronic dehydration in nursing home residents, new diagnostic methods can be established. Therefore, the objectives of this study were (i) to identify risk factors for chronic dehydration diagnosed based on serum osmolality in older nursing home residents in Japan and (ii) to verify the validity of using ultrasound to measure IVC or IVC collapsibility as a method for diagnosing chronic dehydration.

## 2. Materials and Methods

### 2.1. Study Design

This prospective observational study was conducted in nursing homes with the aim of identifying risk factors and related outcomes associated with chronic dehydration. Written informed consent was obtained from all participants. If participants were unable to provide consent, we asked family members to complete and sign the form on their behalf after receiving a full explanation of the study protocol. The protocol was approved by the Ethics Committee of Nagoya University Graduate School of Medicine (approval no. 2019-0012) and adhered to the principles of the Declaration of Helsinki and its later amendments.

### 2.2. Participation Criteria

Residents aged ≥65 years living in 5 nursing homes in Aichi Prefecture, Japan, were recruited and enrolled between May 2019 and March 2020. Inclusion criteria were living in the nursing home for ≥1 week without requiring urgent medical care for an acute illness. Exclusion criteria were intravenous fluid infusion; dialysis; and use of a ventilator, continuous positive airway pressure, or home-oxygen.

### 2.3. Data Collection

#### 2.3.1. Questionnaire

Comprehensive assessments were conducted on the survey date using a structured questionnaire. Nurses in each nursing home completed the questionnaire, and clinicians checked the responses later. The questionnaire contained items on basic information, health status, and functional status such as age, sex, body mass index (BMI), ADL, frailty, swallowing function, nutritional status, comorbidities, constipation, medication, level of care need, history of falling, fracture, and syncope.

BMI was categorized into 3 groups: underweight (<18.5), normal weight (18.5 to <25), and overweight or obese (≥25) [21]. ADL was assessed by the Barthel Index (BI), which comprises 10 categories: eating, transfers, grooming, toileting, bathing, walking, stairs, dressing, bowels, and bladder [22]. BI is scored from 0 to 100, with a higher score indicating greater independence. Frailty was assessed using the Clinical Frailty Scale (CFS), which measures the overall level of fitness or frailty according to 9 categories: 1, very fit; 2, well; 3, managing well; 4, vulnerable; 5, mildly frail; 6, moderately frail; 7, severely frail; 8, very severely frail; and 9, terminally ill [23].

The Dysphagia Severity Scale (DSS) was used to assess swallowing function [24]. The DSS has 7 categories: 1, saliva aspiration; 2, food aspiration; 3, water aspiration; 4, occasional aspiration; 5, oral problems; 6, minimal problems; and 7, normal. Scores 1 to 4 are categorized as the Low DSS group and scores 5 to 7 as the High DSS group. Nutritional status was assessed using the Mini Nutritional Assessment-Short Form (MNA-SF), which is a validated nutritional screening tool [25]. The MNA-SF is scored from 0 to 14, with a score of 0 to 7 indicating malnourished, 8 to 11 risk of malnutrition, and 12 to 14 normal nutritional status.

Comorbidities were assessed using the Charlson Comorbidity Index (CCI) [26]. The CCI is the sum of scores for 19 items related to chronic diseases. To calculate CCI, information about physician-diagnosed comorbidities were obtained from the medical records. Clinical diagnosis of dementia based on Diagnostic and Statistical Manual of Mental Disorders (DSM)-V was independently confirmed by 2 geriatricians. Dementia was diagnosed when there was both obvious cognitive decline and need for help with complex instrumental daily activities. When there was disagreement on the initial assessment, the 2 geriatricians reached a consensus through discussion.

Constipation was assessed based on the resident’s complaint or use of laxatives. Number of medication and use of diuretics were obtained from medical records.

The level of care need, which is categorized into 7 groups under the public long-term care insurance system in Japan [27], consists of support levels 1 and 2 and care levels 1 to 5. Support level 1 indicates the highest level of independence and care level 5 the highest level of dependence. The level of care need was obtained from the medical records.

History of falling, fracture, and syncope in the last 12 months was assessed from residents’ reporting such events or from the medical records.

#### 2.3.2. Blood Samples

Blood tests were taken within 1 month of the survey date. All blood samples were obtained by experienced nurses and packed in ice for transportation to Nagoya Clinical Laboratory. Serum osmolality (measured by freezing point depression), BUN, creatinine, Na, total protein, albumin, triglyceride, glucose, and brain natriuretic peptides (BNP) or N-terminal proBNP (NT-proBNP) were analyzed. All blood data were collected from medical records. Serum osmolality was classified into two groups: euhydration (275 to <295 mOsm/kg) and chronic dehydration (≥295 mOsm/kg) [10]. Estimated glomerular filtration rate (eGFR) was calculated using a formula for the Japanese population: eGFR(ml/min/1.73 m^2^) = 194 × serum Cr^−1.094^ × age^−0.287^ (× 0.739 if female) [28].

#### 2.3.3. IVC Measurement and Physical Examination

The researcher visited each participant and performed IVC measurements and physical examinations on the morning of a predetermined day within a month from the survey date, after the participants were instructed to refrain from eating.

All the examinations were performed by the same geriatrician (M.N.), who had 7 years of experience assessing IVC by ultrasonography and had completed a POCUS training course. All IVC measurements were performed using portable handheld ultrasound devices (Vscan Dual Probe, Vscan Extend; GE Healthcare Japan, Tokyo, Japan) with a 3.5-MHz sector probe. IVC was measured from the long-axis/subxiphoid view at 1.0 to 2.0 cm from the junction with the right atrium in the supine position [29]. Minimum (inspiratory phase) and maximum (expiratory phase) IVC diameters were measured using B-Mode under normal breathing. The IVC collapsibility index (IVC-CI) was calculated from the minimum and maximum IVC diameters as follows: IVC-CI = (IVCmax−IVCmin)/IVCmax.

A physical examination screening was performed before the IVC measurement. Physical examination included blood pressure measurement and signs of dry tongue, dry axilla, poor skin turgor, and prolonged capillary refill time (CRT). Blood pressure was measured in the supine position. Dry tongue was determined as dryness of the tongue or oral mucous membranes on visual assessment. Dry axilla was assessed by palpating under the armpit (dry or wet). Poor skin turgor was determined by pinching the skin over the sternum and observing whether the skin fold returned to its normal position. CRT was determined by compressing the nail of the middle finger and measuring how much time passed before it returned to its normal color; prolonged CRT was defined as >2 s.

### 2.4. Statistical Analysis

Participant characteristics were compared between the euhydration group (serum osmolality, 275 to <295 mOsm/kg) and the chronic dehydration group (serum osmolality, ≥295 mOsm/kg) using the *t*-test or Mann–Whitney U-test for continuous variables and the χ^2^ test or Fisher’s exact test for categorical variables. Next, multivariate logistic regression analysis was performed to examine the association between chronic dehydration and exposures that were significant in the univariate analysis. Serum osmolality ≥295 was applied in the logistic analysis as an objective variable. The relationships between IVC variables were analyzed using Pearson’s or Spearman’s correlation test. In all comparisons, *p* < 0.05 was considered statistically significant. All statistical analyses were performed using SPSS version 26 (IBM Corporation, Chicago, IL, USA).

## 3. Results

A total of 108 residents living in 5 nursing homes were enrolled. Of these, 19 were excluded for the following reasons: no blood samples collected (*n* = 2), no serum osmolality measured (*n* = 7), and serum osmolality <275 mOsm/kg (*n* = 10). Finally, 89 residents were analyzed; their background characteristics are shown in Table 1. Seventy-four residents (83.1%) were classified into the euhydration group and 15 (16.9%) into the chronic dehydration group. We found a significantly higher proportion of dementia, lower eGFR, and higher BMI in the chronic dehydration group.

The results of the multivariate logistic regression analysis to investigate factors associated with chronic dehydration are shown in Table 2. Model 1 (adjusted by age and sex) revealed that chronic dehydration showed associations with dementia, lower eGFR, and higher BMI, whereas Model 2 (adjusted by age, sex, dementia, eGFR, and BMI) showed associations with dementia (odds ratio (OR), 6.290; 95% CI, 1.270–31.154) and higher BMI (OR, 1.471; 95% CI, 1.105–1.958). Model 3 showed that the relationship between chronic dehydration and dementia did not change (OR 8.619, 95%CI, 1.350–55.050) when we included CCI (excluding dementia), BI, DSS, and number of medications as confounding factors. In addition, the association between chronic dehydration and higher BMI was seen only in women (Table 3).

No association was seen between chronic dehydration and IVC or IVC-CI (Table 4). Moreover, there were no significant correlations among IVCmax, IVC-CI, and the parameters BUN/Cr, Na, height, and NT-proBNP (Table 5).

## 4. Discussion

In this study, we investigated risk factors for chronic dehydration diagnosed based on high serum osmolality in nursing home residents in Japan. Moreover, we assessed the validity of using ultrasound to measure IVC or IVC collapsibility as a method for diagnosing chronic dehydration. We found that chronic dehydration showed associations with dementia and higher BMI but not IVC or IVC-CI.

In the current study we did not find significant differences between euhydration and dehydration in BI, DSS, CCI, and medication numbers. However, previous studies identified these factors as possible contributing factors to chronic dehydration in nursing home residents with dementia [30,31,32]. Therefore, these factors could be mediating factors between dementia and hydration.

Insufficient fluid intake leads to water-loss dehydration and may increase blood sodium level and serum osmolality [33]. One study reported that low fluid intake was found in up to 51.3% of long-term care residents with cognitive impairment [34]. Individuals with advanced dementia may forget to drink, forget where they have left their cup, have reduced attention to drinking, not wish to open their mouth, or not wish to swallow [35]. They also may not be able to accurately communicate their thirst to nursing home staff when they feel thirsty [36]. Moreover, nursing home residents with dementia were reported to have nearly twice the risk of anorexia compared with those without dementia [37], which may lead to reduced fluid intake.

In terms of functional impairment, dementia is a strong risk factor for functional decline in nursing home residents [31]. In addition, decline in ADL is associated with severity of dementia in nursing home residents [38]. A previous study reported that low BI scores were associated with low fluid intake [7]. Nursing home residents with dementia that exhibit functional decline also require assistance to maintain sufficient fluid intake. In other words, nursing home residents with dementia and reduced physical function likely have less than adequate water intake.

Dysphagia is common in people with dementia and is more commonly seen in nursing homes. In long-term care facilities, the rate of dysphagia with dementia has been estimated to be as high as 53% [39]. In the advanced stage of dementia, it can become harder to swallow. Dysphagia is thought to be associated with not only malnutrition but also dehydration in those with dementia.

Dementia is commonly associated with other comorbidities such as diabetes, chronic kidney disease, chronic heart failure, and vascular disease. It has been reported that having more than 4 chronic conditions or having diabetes is a risk factor for dehydration [30,32]. Therefore, dehydration may be more likely to occur in nursing home residents with dementia and comorbidities.

Long-term care residents have a high prevalence of multimorbidity and polypharmacy [40], and residents with dementia are prone to polypharmacy as a result of comorbidities [41]. Polypharmacy may reduce appetite, which may in turn lead to malnutrition and dehydration. Older adults may also be susceptible to dehydration resulting from unmonitored use of diuretics [42]. In this study, we examined possible causal relationships based on previous reports, but the association between dementia and chronic dehydration did not change when we analyzed CCI (excluding dementia), BI, DSS, and medication numbers as confounding factors.

However, it is possible that the relationship between dementia and chronic dehydration is bidirectional; that is, dementia might be triggered by chronic dehydration. Dehydration is known to be a risk factor for decreased attention or delirium in nursing home residents [43], and previous studies have also reported that dehydration affects cognitive dysfunction. One study reported that lower hydration status measured by the bioelectrical impedance was associated with slower psychomotor process speed and poor memory task in older adults [44], and another study reported that women aged ≥60 years with higher serum osmolality (≥290 mmol/L) tended to score 3–5 points lower on the Digit Symbol Substitution test compared with women with a serum osmolality of 285–289 mmol/L [45]. This might mean that complicated tasks are more affected by hydration levels. A previous study proposed the hypomolecular hypothesis, which postulates that cognitive impairment is caused by protein misfolding and subsequent aggregation in the brain under conditions of decreased interstitial fluid volume resulting from dehydration. Defective proteins may lead to cognitive impairment by hindering information processing in the biomolecular networks of the brain or through synaptic damage [46]. Patients with cognitive decline such as those with Alzheimer’s disease have a higher risk of dehydration, and dehydration is associated with the risk of dementia [47]. As a result, dehydration could be exacerbated in older people with dementia, which may in turn lead to further cognitive dysfunction.

In this study, we also found chronic dehydration was associated with higher BMI. A previous study reported that dehydration was one of the main causes of weight loss in nursing home residents [48]. However, a more recent investigation of long-term care facilities reported that the proportion of residents with dehydration did not differ by BMI [49]. This result suggests that dehydration may be common in overweight or obese nursing home residents. In fact, one study conducted on a representative sample of older adults in Portugal showed that obesity was associated with higher urine osmolality [50]. This may be explained by the altered fluid distribution in individuals with obesity; that is, obese individuals are significantly more likely to be hypertonic than normal-weight individuals, and thus when osmotic fluid shifts, the increase in extracellular fluid occurs relative to the intracellular fluid [51]. Moreover, another recent study reported that habitual low fluid intake was associated with larger body fluid volume due to overcompensation by the kidneys [52]. That could explain the association between chronic dehydration and higher BMI. In the present study, the prevalence of BMI ≥25 was also significantly higher in the chronic dehydration group than in the euhydration group (35.7% vs. 1.5%). In addition, we found chronic dehydration was associated with higher BMI only in women. This may be explained by the difference in fat ratio between men and women. One previous study reported that women had less total body water compared with men with the same BMI and that the ratio of intracellular water to total body water in women was significantly lower than that in men when BMI increased [53]. Older women with higher BMI are considered to be at increased risk of chronic dehydration.

In the multivariate analysis, lower eGFR did not show significant association with chronic dehydration (Model 2 and Model 3 in Table 2). We performed additional analysis to clarify the association between eGFR and other factors and found a negative correlation between eGFR and BMI (Appendix A). This might be explained by the association between chronic kidney disease and obesity [54].

We could not clarify the effectiveness of measuring IVC or IVC-CI to assess chronic dehydration. Chronic dehydration causes fluid deficits mainly in the intracellular fluid compartment. In other words, increased osmolality shifts intracellular fluid to the extracellular compartment. IVC diameter has been used to predict intravascular volume, but it might be difficult to evaluate intracellular fluid volume using IVC, at least in the early stages of dehydration. Another reason is that, in some cases, it was difficult to examine the IVC anatomically from the long-axis/subxiphoid view. The IVC moves along the long axis with breathing, and thus we might not always be able to accurately assess the IVC diameter. Furthermore, cases in which breathing and sniffing tests could not be performed due to dementia may have affected the IVC or IVC-CI assessment.

In both previous studies and our study, neither physical signs (dry tongue, dry axilla, poor skin turgor, prolonged CRT) nor the BUN/Cr ratio was useful for diagnosing chronic dehydration.

This study produced some important findings, but several limitations should be considered. First, we defined chronic dehydration as serum osmolality ≥295 mOsm/kg that investigated water-loss dehydration, but not other forms of dehydration such as water-and-solute-loss dehydration, which is isotonic or hypotonic due to the greater loss of solutes than water. In other words, we might have misclassified this type of dehydration. In addition, regarding the cut-off points for chronic dehydration, some studies defined serum osmolality 295–300 as impending dehydration, and >300 as current dehydration [10,30]. However, there has been no clear standards for serum osmolality for chronic dehydration, and a previous study used serum osmolality >295 as chronic dehydration [55]. In order to establish effective strategies for early intervention, we classified serum osmolality ≥295 as chronic dehydration in the present study. Second, CCI was relatively low for the nursing home residents and might have been underestimated. However, given that the medical information kept by the nursing homes was limited, we evaluated CCI by methods commonly used in clinical practice. Third, we were unable to unify the date and time of blood collection, so blood samples were collected at different times of day and not necessarily when the participants were in a fasting state. Recent food consumption or hormone secretion may influence serum osmolality. However, the best timing for serum osmolality measurement has been controversial. Indeed, a previous study on dehydration used serum osmolality in non-fasting venous blood [30]. Fourth, we were unable to examine the effect of valvular disease on the assessment of IVC or IVC-CI because we did not evaluate cardiac function at the time of IVC measurement. Evaluation of cardiac function is time-consuming and requires advanced skills, so we measured IVC as a simpler and more practical approach. Lastly, because of the relatively small sample size, we could not perform a more detailed multivariate analysis to assess dehydration.

In conclusion, associations were found between chronic dehydration and dementia or higher BMI in Japanese nursing home residents. IVC and IVC-CI were not useful for diagnosing chronic dehydration. Cognitive function and body weight of residents should be considered when establishing a strategy for preventing chronic dehydration in nursing homes. Given that no method for the early detection of chronic dehydration has been established yet, further investigation of the factors contributing to chronic dehydration is required.

## Figures and Tables

**Table 1 nutrients-12-03562-t001:** Participant characteristics.

	Total	Euhydration	Dehydration	*p*-Value
Participants	89	74	15	
Age, years	87.8 ±6.4	87.5 ± 6.4	89.3 ± 6.2	0.33
Male, *n* (%)	15 (16.9%)	12 (16.2%)	3 (20.0%)	0.71
Level of care need (*n* = 85)				0.26
Support level, *n* (%)	15 (17.6%)	11 (15.5%)	4 (28.6%)	
Care level, *n* (%)	70 (82.4%)	60 (84.5%)	10 (71.4%)	
DSS				1.00
Low DSS, *n* (%)	17 (19.1%)	14 (18.9%)	3 (20.0%)	
High DSS, *n* (%)	72 (80.9%)	60 (81.1%)	12 (80.0%)	
CCI	1.8 ± 1.4	1.7 ± 1.3	2.1 ± 1.9	0.27
Dementia, *n* (%)	50 (56.2%)	38 (51.4%)	12 (80.0%)	0.049
Heart failure, *n* (%)	16 (18.0%)	13 (17.6%)	3 (20.0%)	0.73
Diabetes mellitus, *n* (%)	10 (11.2%)	8 (10.8%)	2 (13.3%)	0.67
eGFR, mL/min/1.73 m^2^	62.0 ± 24.3	64.4 ± 24.9	50.1 ±17.5	0.036
History of falling, *n* (%) (*n* = 79)	27 (34.2%)	23 (34.8%)	4 (30.8%)	1.00
History of fracture, *n* (%) (*n* = 85)	8 (9.4%)	7 (9.9%)	1 (7.1%)	1.00
History of syncope, *n* (%) (*n* = 85)	2 (2.4%)	1 (1.4%)	1 (7.1%)	0.30
History of hospitalization, *n* (%) (*n* = 87)	10 (11.5%)	9 (12.3%)	1 (7.1%)	1.00
Constipation, *n* (%) (*n* = 85)	62 (72.9%)	50 (70.4%)	12 (85.7%)	0.33
CFS, median (quartile) (*n* = 88)	6 (4-7)	7 (4-7)	5 (3-7)	0.078
Barthel Index (*n* = 87)	49.8±33.3	49.3 ± 33.0	52.5±36.3	0.75
Diuretics use, *n* (%)	24 (27.0%)	18 (24.3%)	6 (40.0%)	0.21
Medication numbers	6.7 ± 3.3	6.5 ± 3.1	7. 5± 3.9	0.30
BMI, kg/m^2^ (*n* = 81)	20.9 ± 3.1	20.4 ± 2.9	23.4 ± 3.1	<0.01
BMI ≥ 25, *n* (%) (*n* = 81)	6 (7.4%)	1 (1.5%)	5 (35.7%)	<0.01
MNA-SF (*n* = 82)	8.8 ± 2.6	8.6 ± 2.5	10.0 ± 2.4	0.059
Dry tongue, *n* (%) (*n* = 88)	29 (33.0%)	24 (32.9%)	5 (33.3%)	0.97
Dry axilla, *n* (%)	16 (18.0%)	13 (17.6%)	3 (20.0%)	0.73
Poor skin turgor, *n* (%)	20 (22.5%)	17 (23.0%)	3 (20.0%)	1.00
Prolonged CRT, *n* (%) (*n* = 87)	22 (25.3%)	20 (27.8%)	2 (13.3%)	0.34
BUN/Cr ratio	24.5 (8.8)	24.0 ± 8.4	27.0 ± 10.8	0.24
Serum osmolality, mOsm/kg	288.5 ± 6.1	286.7 ± 4.8	297.5 ± 2.7	
Albumin, g/dL	3.5 ± 0.5	3.5 ± 0.5	3.7 ± 0.4	0.15

Data presented as mean ± SD. DSS, Dysphagia Severity Scale; CCI, Charlson Comorbidity Index; eGFR, estimated glomerular filtration rate; CFS, Clinical Frailty Scale; BMI, body mass index; MNA-SF, Mini Nutritional Assessment-Short Form; Extend CRT, extend capillary refill time; BUN/Cr, blood urea nitrogen-to-creatinine ratio.

**Table 2 nutrients-12-03562-t002:** Factors associated with chronic dehydration in multivariate logistic regression analysis.

	Model 1		Model 2		Model 3	
	Odds ratio (95% CI)	*p*-value	Odds ratio (95% CI)	*p*-value	Odds ratio (95% CI)	*p*-value
Dementia	4.271 (1.080–16.891)	0.038	6.290 (1.270–31.154)	0.024	8.619 (1.350–55.050)	0.023
eGFR	0.965 (0.933–0.999)	0.043	0.974 (0.923–1.028)	0.344	0.986 (0.932–1.045)	0.642
BMI	1.435 (1.128–1.827)	<0.01	1.471 (1.105–1.958)	<0.01	1.492 (1.086–2.048)	0.013

Model 1 was adjusted by age and sex; Model 2 was adjusted by age, sex, dementia, eGFR, and BMI; Model 3 was adjusted by age, sex, dementia, eGFR, BMI, CCI (excluding dementia), BI, DSS, and medication numbers. eGFR, estimated glomerular filtration rate; BMI, Body mass index; CCI, Charlson Comorbidity Index; BI, Barthel Index; DSS, Dysphagia Severity Scale; CI: confidential interval.

**Table 3 nutrients-12-03562-t003:** BMI differences between euhydration and dehydration stratified by sex.

	Euhydration	Dehydration	*p*-Value
BMI, kg/m^2^			
Male (*n* = 15)	21.0 ± 2.9 (*n* = 12)	22.0 ± 1.5 (*n* = 3)	0.57
Female (*n* = 66)	20.3 ± 2.9 (*n* = 55)	23.8 ± 3.4 (*n* = 11)	<0.01

Data presented as mean ± SD (*t*-test). BMI, body mass index.

**Table 4 nutrients-12-03562-t004:** IVC/IVC-CI differences between euhydration and dehydration.

	Total	Euhydration	Dehydration	*p*-Value
IVC diameter, mm	10.5 (8.1–12.6) (*n* = 83)	10.0 (8.1–12.6) (*n* = 69)	11.5 (9.4–13.2) (*n* = 14)	0.22
IVC CI, %	42.8 ± 13.4 (*n* = 80)	42.9 ± 13.0 (*n* = 67)	42.0 ± 16.0 (*n* = 13)	0.83

Data presented as median (quartile) (Mann–Whitney U-test) or mean ± SD (*t*-test). IVC, inferior vena cava; IVC-CI, IVC collapsibility index.

**Table 5 nutrients-12-03562-t005:** Correlation coefficient between IVC and related parameters.

	IVC	IVC-CI
Serum Osmolality	0.215	−0.035
BUN/Cr ratio (*n* = 83)	0.062	−0.002
Na (*n* = 83)	0.215	0.001
Height (*n* = 81)	0.168	−0.122
NT-proBNP (*n* = 78)	0.128	−0.18

NT-proBNP, N-terminal proBNP; BUN/Cr, BUN/Cr, blood urea nitrogen-to-creatinine ratio.

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
