# Peer review of "Chronic Dehydration in Nursing Home Residents"

_nutrients, 2020, doi:10.3390/nu12113562_

Round 1
Reviewer 1 Report
Thank you for allowing me to review the article about chronic dehydration by Nagae et al. The group studies what distinguishes nursing home residents with a serum osmolality being higher or lower than 295 mosmol/kg. They new outcome measure was the collapsibility of the inferior vena cava, which is considered to be a modestly useful method to capture intravascular volume loss (hypovolemia). Here, it did not prove to be useful, which I think is the key result of the study.
Questions and notes:
- How long was the time lapse between the collection of the different parameters (days, weeks, years?)
- The literature usually sets "dehydration" to equal serum osmolality > 300 mosmol/kg because 295 is the mean normal value. Here, the mean value seems to be lower. Comment?
- The authors claim that insufficient fluid intake will lead to intracellular dehydration. This is a common notion, but actual measurements suggest otherwise (see article by Hahn RG in Eur J Nutr, which might even explain why dehydrated patients had a higher BMI).
- The significant difference in eGFR between euhydrated/dehydrated shown in Table 1 is lost in the multivariate analysis. This points at some covariance between eGFR and another factor. Comment?
- You may also conclude that your physical tests (dry tongue etc.) did not indicate dehydration.
- Statistics and references seem to have been carefully carried out.
- The experience from this institution is that poor intake of water results in renal conservation of water, which is conveniently detected by urine sampling. Hyperosmolality develops when poor renal function is added to the picture. Your data does not seem to clearly support that view.
Author Response
Thank you for allowing me to review the article about chronic dehydration by Nagae et al. The group studies what distinguishes nursing home residents with a serum osmolality being higher or lower than 295 mosmol/kg. They new outcome measure was the collapsibility of the inferior vena cava, which is considered to be a modestly useful method to capture intravascular volume loss (hypovolemia). Here, it did not prove to be useful, which I think is the key result of the study.
Thank you for your encouraging and productive comments on our manuscript. We agree with all your advices, and revised our manuscript accordingly. Please see below for specific points.
Red color on revise1 indicates the parts that we changed according to your comments.
1. How long was the time lapse between the collection of the different parameters (days, weeks, years?)
We performed blood tests, IVC measurement and physical examination within 1 month from the survey date when the questionnaire was filled in.
We added the description in p3 line 112, and p4 line 157.
2. The literature usually sets "dehydration" to equal serum osmolality > 300 mosmol/kg because 295 is the mean normal value. Here, the mean value seems to be lower. Comment?
We agree with your comment. Some studies defined serum osmolality 295-300 as impending dehydration, and >300 as current dehydration as you mentioned.
However, there has been no clear standards for serum osmolality for chronic dehydration, and a previous study used serum osmolality 295 as one of the cut-off points for chronic dehydration (Gerontology 2007; 53: 179-183) . Moreover, in order to establish effective strategies for early intervention, we classified serum osmolality ≥ 295 as chronic dehydration in the present study.
We added the description in p8 line 358-366.
3. The authors claim that insufficient fluid intake will lead to intracellular dehydration. This is a common notion, but actual measurements suggest otherwise (see article by Hahn RG in Eur J Nutr, which might even explain why dehydrated patients had a higher BMI).
Thank you for your educational comment.
The article suggested that habitual low fluid intake was associated with larger body fluid volume due to overcompensation by the kidneys. That could explain the association between chronic dehydration and higher BMI. We decide to describe some discussion citing this article (p8 line 331-333).
4. The significant difference in eGFR between euhydrated/dehydrated shown in Table 1 is lost in the multivariate analysis. This points at some covariance between eGFR and another factor. Comment? 7. The experience from this institution is that poor intake of water results in renal conservation of water, which is conveniently detected by urine sampling. Hyperosmolality develops when poor renal function is added to the picture. Your data does not seem to clearly support that view.
Thank you for your valuable comments.
We performed additional analysis to clarify the association between eGFR and another factor, and found negative correlation between eGFR and BMI (we added in supplement 1). This might be explained by the association between chronic kidney disease and obesity (BMC Geriatr 2020; 20: 366).
We are sorry but we didn’t perform urine tests. In this study, eGFR was affected by confounding factors, and the small sample size might also affect the analysis.
We added the description in p8 line 341-345.
5. You may also conclude that your physical tests (dry tongue etc.) did not indicate dehydration
We agree with your comment. We described them in p8 line 355-356.
6. Statistics and references seem to have been carefully carried out.
We appreciate your comment very much.
Reviewer 2 Report
The paper by Nagae and colleagues investigates through a prospective observational study the potential risk factors for chronic dehydration and verify the useful of IVC or IVC-CI in the diagnosis of chronic dehydration.
The study is interesting and the methods are well described; moreover, the chronic dehydration in the elderly is a growing area of research. I have only a few minor issues to enlighten.
MINOR CONCERNS
- As correctly stated by the authors, the study population is somewhat limited in terms of sample size (especially a small proportion of participants with dehydration and male participant). I believe that the authors analyzed the data appropriately, but probably due to this issue they did not find many significant dependencies. This concern has been enlightened by the authors as a limitation, thus I do not have additional comments and I do not require modifications to their analysis.
- The tables could be somewhat better described (no information which test was used to determine the p-value; no the n number in tables 3 and 4 in headline).
- Why was there no correlation analysis for IVC variables and osmolality?
- In the first part of the discussion, there is no clear discussion of own results. In my opinion, a few sentences linking own results with those discussed in the discussion are missing (dysphagia, comorbidities, polypharmalogy).
- Line 260-261, cited by the authors item in the literature (44) concerns adults (21-46 y), not the older adults. I suggests citing other studies in this area, although the research results are scarce and inconclusive.
- I will not agree that the limitation of the study was to not perform MMSE or CDR. The authors used clinical diagnosis, and MMSE is only a screening method.
Author Response
The paper by Nagae and colleagues investigates through a prospective observational study the potential risk factors for chronic dehydration and verify the useful of IVC or IVC-CI in the diagnosis of chronic dehydration.
The study is interesting and the methods are well described; moreover, the chronic dehydration in the elderly is a growing area of research. I have only a few minor issues to enlighten.
Thank you for your encouraging and productive comments on our manuscript. We agree with all your advices, and revised our manuscript. Please see below specific points.
Blue color on revise1 indicates the parts that we changed according to your comments.
-As correctly stated by the authors, the study population is somewhat limited in terms of sample size (especially a small proportion of participants with dehydration and male participant). I believe that the authors analyzed the data appropriately, but probably due to this issue they did not find many significant dependencies. This concern has been enlightened by the authors as a limitation, thus I do not have additional comments and I do not require modifications to their analysis.
We appreciate your comment very much.
We recognize that this research has several limitations due to the small sample size.
-The tables could be somewhat better described (no information which test was used to determine the p-value; no the n number in tables 3 and 4 in headline).
We agree with your comment.
We added or corrected descriptions in Table 3 and Table 4.
- Why was there no correlation analysis for IVC variables and osmolality?
Thank you for your comment.
We performed correlation analysis for IVC, IVC-CI and serum osmolality, and added in Table5.
- In the first part of the discussion, there is no clear discussion of own results. In my opinion, a few sentences linking own results with those discussed in the discussion are missing (dysphagia, comorbidities, polypharmalogy).
Thank you for your comment.
In the current study we did not find significant differences between euhydration and dehydration in BI, DSS, CCI, and medication numbers. However, previous studies identified these factors as possible contributing factors to chronic dehydration in nursing home residents with dementia. We added and corrected the description p6 line 227-231.
-Line 260-261, cited by the authors item in the literature (44) concerns adults (21-46 y), not the older adults. I suggests citing other studies in this area, although the research results are scarce and inconclusive.
We apologize our mistakes to cite incorrect literature.
We cited another study in older adults, and added description in p7 line 285-287 and references [44].
-I will not agree that the limitation of the study was to not perform MMSE or CDR. The authors used clinical diagnosis, and MMSE is only a screening method.
Thank you for your comment.
We deleted the description what you pointed out (in p8 line 366).